# Current State of Anthrax Vaccines and Key R&D Gaps Moving Forward

**DOI:** 10.3390/microorganisms8050651

**Published:** 2020-04-29

**Authors:** Adam Clark, Daniel N. Wolfe

**Affiliations:** Biomedical Advanced Research and Development Authority, Assistant Secretary for Preparedness and Response, Department of Health and Human Services, Washington, DC 20201, USA; adam.clark@hhs.gov

**Keywords:** anthrax, Bacillus, anthracis, vaccine

## Abstract

A licensed anthrax vaccine has been available for pre-exposure prophylaxis in the United States since 1970, and it was approved for use as a post-exposure prophylaxis, in combination with antibiotic treatment, in 2015. A variety of other vaccines are available in other nations, approved under various regulatory frameworks. However, investments in anthrax vaccines continue due to the severity of the threat posed by this bacterium, as both a naturally occurring pathogen and the potential for use as a bioweapon. In this review, we will capture the current landscape of anthrax vaccine development, focusing on those lead candidates in clinical development. Although approved products are available, a robust pipeline of candidate vaccines are still in development to try to address some of the key research gaps in the anthrax vaccine field. We will then highlight some of the most pressing needs in terms of anthrax vaccine research.

## 1. Introduction

Anthrax is an acute infectious disease caused by the Gram-positive, spore-forming bacterium *Bacillus anthracis*, which can cause human disease via gastrointestinal, cutaneous, or inhalation (pulmonary) routes. In North America, anthrax infections are rare in humans, normally associated with contact with infected animals or exposure to infected animal products. Cutaneous anthrax accounts for more than 95% of human cases and can be effectively treated with antibiotics [1]. Inhalation anthrax, however, has a mortality rate of around 90% when left untreated and is capable of being weaponized as a biological agent [2]. The National Institute of Allergy and Infectious Disease (NIAID) lists *B. anthracis* as a Category A priority pathogen, which poses the highest risk to national security; U.S. Department of Homeland Security identified anthrax as a threat to national security, issuing a material threat determination in 2004.

Given the lethality of anthrax disease, especially via the inhalational route of exposure, and the potential use as an agent of bioterrorism, vaccines against anthrax have been developed and approved for use. In June 1993, the use of anthrax as an agent of bioterrorism was attempted in Japan. Fortunately, much of the bacterial culture grew weakly when tested and lacked amplification of the pXO2-at marker, suggesting that it was the Sterne strain used in veterinary vaccines [3]. However, the 2001 Amerithrax attacks highlighted the damage that can be done by anthrax, even on a small scale. In this case, letters filled with anthrax were mailed and opened, demonstrating for the first time the use of anthrax as a bioweapon on the U.S. population and inciting terror and anxiety amongst the public. Twenty-two individuals became infected with anthrax, resulting in five fatal cases and an estimated $177 million in medical costs [4]. Beyond the medical tolls, an estimated $320 million was also needed for decontamination efforts [5]. A human anthrax vaccine was initially developed in the United States in the 1950s and was originally approved for use in 1970, but there had been little progress in the development of new medical countermeasures for general use, or use as a post-exposure prophylaxis at the time of the 2001 Amerithrax attacks.

Renewed investments in anthrax medical countermeasures followed the 2001 attack. A 2002 report by the Institute of Medicine highlighted some of the key recommendations for anthrax vaccines moving forward [6]. In the nearly two decades since, substantial progress has been made with the approved anthrax vaccine as well as a suite of next-generation vaccine candidates. Multiple funding agencies and sponsors have pursued efforts to improve the vaccine schedule and dosing strategies. The collective preparedness posture for the United States now includes a vaccine that is also licensed for post-exposure prophylaxis when used in combination with antibiotics, as of November 2015 [7].

Anthrax infection pathology and virulence are driven in large part by two sets of genes. The pXO1 plasmid carries the genes for protective antigen (PA), lethal factor (LF), and edema factor (EF). PA binds to LF and EF to form lethal toxin (LT) and edema toxin (ET), respectively. Binding by PA enables the entry of LF and EF into cells, resulting in cellular toxicity, and contributing to the lethality of the disease [8]. The pXO2 plasmid carries genes for capsule production and regulation, and also plays a key role in anthrax disease [9]. However, the central role of PA in the toxic effects of anthrax infections have resulted in anti-PA and toxin-neutralizing antibodies being important correlates of protection, as well as PA being a central antigen in recent vaccine efforts. Here, we will summarize the current state of anthrax vaccines and discuss some key gaps that, if filled, would further improve our preparedness for future events involving anthrax exposures.

## 2. The Current State of Anthrax Vaccines

### 2.1. Vaccines Currently in Use

The vaccine BioThrax^®^ (also known as AVA or Anthrax vaccine adsorbed) was initially approved for pre-exposure prophylaxis in 1970. As a pre-exposure prophylaxis, the current schedule for BioThrax^®^ involves a three-dose primary series of intramuscular injections at zero, one, and six months with booster vaccinations required at 6 and 12 months after the primary series. Annual boosts are required thereafter [7]. Clinical trials have been conducted over the past two decades to inform the current dosing schedule and boosting strategy [10,11]. The indication for pre-exposure prophylaxis specifies those at high risk of exposure; military personnel, certain laboratory workers, and individuals handling animals (such as veterinarians) may have access to the vaccine. A Phase 3 study published by Hopkins et al. demonstrated the potential utility as a post-exposure prophylaxis when administered subcutaneously at zero, two, and four weeks [12]. Previous nonclinical data indicated that the vaccine was effective as a post-exposure prophylaxis when administered in combination with antibiotics [13]. The current recommendation for post-exposure prophylaxis includes this regimen when combined with 60 days of appropriate antimicrobial prophylaxis.

Biothrax^®^ is manufactured from the cell-free extracts of an avirulent strain of *B. anthracis* and is capable of generating antibodies against the PA protein to neutralize the anthrax toxins, providing protection in a nonclinical anthrax challenge model [14]. Biothrax^®^ is administered through intramuscular injection for pre-exposure prophylaxis and subcutaneously for post-exposure prophylaxis, with the most common adverse reactions being tenderness, pain, erythema, edema, muscle aches, fatigue, and headaches [15]. Expanding the licensure of BioThrax^®^ involved the FDA Animal Rule. As part of the licensure pathway, an understanding of immune correlates was critical to predicting efficacy in humans based on nonclinical studies. Quantification of toxin-neutralizing antibodies in a toxin neutralization assay (TNA) proved to be a species-neutral correlate of protection in both rabbits and nonhuman primates [16]. Although not a functional assay, ELISA was used to quantify anti-PA IgG and also correlated with protection in a nonhuman primate infection model [17].

For use as a post-exposure prophylactic, BioThrax^®^ would be subcutaneously administered to individuals at zero, two, and four weeks, following suspected or confirmed *B. anthracis* exposure, and administered in conjunction with antimicrobial therapy for 60 days. The approval of a post-exposure indication for BioThrax^®^ greatly enhanced U.S. preparedness and response, however, the three-dose regimen and 60 days of concomitant antibiotic therapy are challenging from an operational perspective as well as for patients to adhere to the recommendations. Following the 2001 anthrax attacks, only 44% of potentially exposed individuals took the 60-day antibiotic course [18]. Similarly, in 2014 a laboratory at the Centers for Disease Control and Prevention (CDC) had an incident that potentially exposed 42 people to aerosolized *B. anthracis*. A post-exposure treatment with BioThrax^®^ and antibiotics was recommended for these individuals. However, 67% of individuals declined to complete the three-dose vaccine regimen, and only 33% reported completing the 60-day antibiotic course [19].

Anthrax vaccine precipitated (AVP) is licensed for use in the United Kingdom. AVP is a cell-free filtrate of the *B. anthracis* Sterne 34F2 strain, precipitated with alum. AVP contains the three major toxin components of anthrax, with approximately 7.9 µg/mL of PA, 1.9 µg/mL of LF, and low but detectable amounts of EF [20]. Like the pre-exposure prophylaxis regimen for BioThrax^®^, AVP is also administered via intramuscular injection. Doses are given at zero, three, six, and 32 weeks, with annual boosts thereafter [21].

A recent clinical trial compared antibody responses to AVP in comparison to BioThrax^®^. While these vaccines elicited similar levels of anti-PA IgG antibodies and ET-neutralizing antibodies, AVP elicited higher titers of anti-EF IgG antibodies [21]. Anti-LF IgG titers were also higher in AVP recipients; the anti-LF antibodies appeared to also impact neutralizing activity as the ED_50_ values to neutralize LT were also higher in AVP recipients [22].

A live-attenuated anthrax vaccine is approved for human vaccination in Russia, via cutaneous and subcutaneous administration. The original formulation of this vaccine was developed in the 1940s and consisted of live dry spores of two different nonencapsulated *B. anthracis* variants [23]. The current formulation now uses one of the strains in combination with PA adsorbed on to aluminum hydroxide. This formulation requires annual subcutaneous injections for three years, with boosts every two years thereafter [23]. A live attenuated anthrax vaccine is also available for human use in China as a suspension of the attenuated strain A16R [24].

### 2.2. Vaccines in Development

Although licensed vaccines are available, investments are still being made in next-generation anthrax vaccines that may produce safer and more effective options. There are several candidates in preclinical development, but this summary will cover those candidates in clinical development over the past five years. Several additional vaccine candidates continue to be evaluated in preclinical stages of development, but the focus on the current pipeline will be limited to allow a focused discussion of gaps in anthrax vaccine programs.

AV7909 (NuThrax^TM^) is a next-generation anthrax vaccine composed of the currently licensed BioThrax^®^ adjuvanted with the CPG 7909 compound, a Toll-like receptor 9 (TLR9) agonist. CPG 7909 is an oligodeoxynucleotide that has been shown to be capable of activating B-cells and enhancing vaccine immunogenicity [25]. A similar CPG molecule is currently used in the Hepatitis B vaccine Heplisav-B^®^. AV7909 and is being developed for post-exposure prophylaxis in combination with antibiotics as a two-dose vaccine. Animal studies in guinea pigs and nonhuman primates have shown complete protection against an anthrax challenge when the vaccine was administered in a two-dose setting (0 and 28 days with a challenge on day 70) [26]. Clinical studies comparing AV7909 with BioThrax^®^ using a 2-week vaccination schedule (0 and 14 days) demonstrated AV7909 rapidly produced anthrax toxin neutralizing antibody titers and had a significantly higher magnitude of response compared to BioThrax^®^ [27]. A Phase 2 clinical trial expanded on the immunogenicity data, further supporting the potential to use AV7909, with a two-dose regimen at days 0 and 14 [28]. AV7909 is currently under investigation in a Phase 2 drug-to-drug interaction study and a Phase 3 lot-to-lot consistency study (clinicaltrials.gov NCT04067011 and NCT03877926).

A variety of different vaccines based on recombinant PA (rPA) have been developed in efforts to provide vaccines that may be easier to manufacture and safer to administer. Friedlander et al. reviewed rPA based vaccines as of 2009 [29], and here we will focus on rPA based vaccines in clinical development over the past five years. A variety of rPA based vaccine candidates have been evaluated in clinical trials over the past five years. These candidates include rPA produced in plants (clinicaltrials.gov NCT02239172), or bacterial production platforms (clinicaltrials.gov NCT01624532, NCT04148118, and NCT02655549). Each of these studies utilized at least two doses. It is proposed that the candidate BW-1010 would be administered intranasally, while the others would be administered via intramuscular injection.

Two additional vaccine candidates based on PA have attempted to use viral vectors to deliver the antigen. Adenovirus serotype 4 has been used to express PA and was evaluated in a Phase 1 clinical trial (clinicaltrials.gov NCT01979406). More recently, AdVAV is a replication-deficient adenovirus type 5 vectored PA that was administered via intranasal dosing and protected against lethal aerosol exposure in a rabbit challenge model [30] and has since progressed into clinical development. AdVAV would be administered via intranasal exposure.

## 3. Key Research and Development Gaps for Anthrax Vaccines

Vaccines are available to protect against anthrax, both as pre- and post-exposure prophylaxis, but there are research and development gaps that remain in the anthrax vaccine space. There are still gaps in our understanding of how the currently licensed vaccines could be utilized, and it is important to take into account the other antibiotics and antitoxins that are available for use as well. For example, BioThrax^®^ has a label indication that includes adults aged 18–65 per its package insert, and a more robust immune response is elicited in younger subjects relative to older subjects. It is currently unclear if a protective immune response would be elicited by the licensed vaccines in adults over the age of 65. To address this question, a Phase 2 clinical trial was conducted in which BioThrax^®^ was evaluated in adults aged 18–50 compared to those over the age of 65. The next-generation vaccine AV7909 was also included to assess whether or not the added adjuvant would be needed to induce a protective immune response (clinicaltrials.gov NCT03518125). Data from this trial will be available later this year.

Other special populations will continue to be outstanding questions in the near future. An initial meta-analysis was undertaken to assess potential issues in trying to protect pediatric subjects using BioThrax^®^. The analyses compared safety and immunogenicity from previous clinical trials in adult subjects aged 18–20 to those aged 21–29. The findings suggested that safety and immunogenicity profiles were similar, and dosing adolescent subjects may not present any additional safety risks [31]. That being said, it will be difficult to fully address the use of any licensed anthrax vaccine in pediatric or adolescent subjects ahead of an emergency event, as use in these populations is highly unlikely in the absence of potential benefit. In addition to pediatric subjects, pregnant and lactating women and immune-compromised subjects will remain to be subsets of the population, with limited data to support using anthrax vaccines in the absence of an emergency.

Improvements to the dosing amounts and schedules may also be desirable. For both AVA and AVP, a primary series of three doses are required. In the case of pre-exposure prophylaxis, the schedule for AVA has been evaluated in clinical trials, and it is now recommended that boosting could occur every three years as opposed to annually in those not at high risk of exposure [32]. Phase 3 development of AV7909 may ultimately reduce the number of required doses to two for post-exposure prophylaxis for a product licensed in the United States, but the potential for pre-exposure prophylaxis to inform the duration of protection and a boosting schedule would require additional clinical data. Changes to the formulation of the currently licensed vaccines may also be desirable. For example, lyophilized formulations may dramatically increase the shelf-lives of these vaccines, which would, in turn, enable a more cost-effective approach to maintaining a sufficient inventory of vaccines.

Improving the operational aspects of licensed products may be an achievable goal in the near-term. In the longer term, ten years or more into the future, it would be desirable to have revolutionary advances in anthrax vaccine capabilities. An anthrax vaccine that would be effective in a single dose would be highly desirable. In a response scenario, a single dose vaccine would be ideal to negate the need for repeated visits. If a single-dose vaccine was also able to provide more rapid onset to protection, that would also be a more ideal post-exposure prophylaxis in that it may reduce the duration that concomitant antibiotic treatment is required. Alternative delivery strategies would also be desirable in a next-generation vaccine candidate. While intramuscular or subcutaneous vaccine administration is feasible, alternative approaches using patch- or microneedle-based delivery, intranasal administration, or oral dosing would also facilitate an emergency response.

## 4. Summary

Multiple anthrax vaccines are licensed and available for use in their respective nations if needed. These vaccines, combined with antibiotics and antitoxins, provide a high level of national preparedness if a major anthrax event were to occur. That said, there are still gaps that remain with the current products, and longer-term gaps that, if filled, could provide more optimal solutions. It is imperative that the anthrax vaccine community continues to support programs with the licensed vaccines to answer key operational questions. In parallel, investments are warranted in early-stage candidates that may enable revolutionary advances, such as single-dose vaccines, or other revolutionary improvements over the current concept of operations. A continued focus on improving our medical countermeasure capabilities for anthrax will ensure the optimal use of licensed vaccines and potentially provide game-changing solutions in the years to come.

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
