# Peer review of "Current State of Anthrax Vaccines and Key R&D Gaps Moving Forward"

_microorganisms, 2020, doi:10.3390/microorganisms8050651_

Round 1

Reviewer 1 Report

Major comments:

Line 75, right after reference [12]: With respect to the post-exposure therapeutic use of the vaccine, it may make sense to cite the 1993 paper by Art Friedlander et al. “Postexposure Prophylaxis against Experimental Inhalation Anthrax”. While the current review is rightfully focused on human clinical data, in light of the animal efficacy rule the monkey data published by Friedlander is relevant and explains why the post-exposure vaccine is given in conjunction with antibiotics and not as a monotherapy. The authors of 1993 paper showed that in the absence of any antibiotics the vaccine is unable to protect animals when administered post-exposure.

Line 156: Section on Gaps. Would it make sense to test the efficacy of the vaccine post-exposure in combination with the approved antitoxins, such as Raxibacumab, Anthim, or Anthrasil? The use of these antitoxins could be used in addition to antibiotics and vaccines. Conveniently, Emergent Biosolutions that owns BioThrax also owns Raxibacumab and Anthrasil.

Minot comments:

Line 90: Italicize B. anthracis.

Line 97: …exposed 42 people to aerosolized B. anthracis (not anthrax).

Line 112 vs for example line 46: be consistent with whether to capitalize Anthrax or not.

Line 294: Reference #28: The names of the authors are shown only as initials.

Author Response

We would like to thank Reviewer 1 for the constructive comments. The comments are pasted below with our responses bolded.

Major comments:

Line 75, right after reference [12]: With respect to the post-exposure therapeutic use of the vaccine, it may make sense to cite the 1993 paper by Art Friedlander et al. “Postexposure Prophylaxis against Experimental Inhalation Anthrax”. While the current review is rightfully focused on human clinical data, in light of the animal efficacy rule the monkey data published by Friedlander is relevant and explains why the post-exposure vaccine is given in conjunction with antibiotics and not as a monotherapy. The authors of 1993 paper showed that in the absence of any antibiotics the vaccine is unable to protect animals when administered post-exposure. We concur, that reference is highly relevant and has been added, with the following sentence “Previous nonclinical data indicated that the vaccine was effective as a post-exposure prophylaxis when administered in combination with antibiotics.”

Line 156: Section on Gaps. Would it make sense to test the efficacy of the vaccine post-exposure in combination with the approved antitoxins, such as Raxibacumab, Anthim, or Anthrasil? The use of these antitoxins could be used in addition to antibiotics and vaccines. Conveniently, Emergent Biosolutions that owns BioThrax also owns Raxibacumab and Anthrasil. We concur. It is important to consider all of the approved countermeasures when discussing the operational studies that would be informative. We have added the statement, “…and it is important to take into account the other antibiotics and antitoxins that are available for use as well.” More specifics could be added if desired, but that would require a more extensive discussion and more space for the manuscript.

Minot comments:

Line 90: Italicize B. anthracis. Thank you for catching, this has been corrected.

Line 97: …exposed 42 people to aerosolized B. anthracis (not anthrax). We concur, that is the appropriate way to say it and revised as such.

Line 112 vs for example line 46: be consistent with whether to capitalize Anthrax or not. Thank you, all are now not capitalized.

Line 294: Reference #28: The names of the authors are shown only as initials. This has been corrected (now reference #29). The authors are Friedlander AM and Little SF.

Reviewer 2 Report

The authors review ongoing work to improve anthrax vaccines.  They specifically exclude pre-clinical work and emphasize ongoing clinical trials.  The text is well written, and they have cited the key clinical trials. 

The authors recommend “revolutionary improvements” in future anthrax vaccines.  This is most likely to result from inclusion of additional antigens in future vaccines.  Examples might be spore coat proteins, HtrA, poly-glutamate, MntA, etc.   Yet no mention is made of including additional antigens.

Formatting of references is inconsistent.  It seems this was not done with a bibliographic manager, e.g. Endnote, or the program was not configured to handle documents from sources other than journals. 

Ref. 8 and 9 are book chapters.  Reviews in more easily accessible journals would be of more value to readers.

Ref. 8:   “membrane”, not “membrain”

Line 90:   intalize B. anthracis

Line 93.  Antibiotic, not antibody.

Line 233:   anthrax

Line 262:   clinical

 Line 271:   spacing lost

Author Response

We would like to thank Reviewer 2 for the constructive comments. Our responses are bolded below.

The authors review ongoing work to improve anthrax vaccines.  They specifically exclude pre-clinical work and emphasize ongoing clinical trials.  The text is well written, and they have cited the key clinical trials. 

The authors recommend “revolutionary improvements” in future anthrax vaccines.  This is most likely to result from inclusion of additional antigens in future vaccines.  Examples might be spore coat proteins, HtrA, poly-glutamate, MntA, etc.   Yet no mention is made of including additional antigens. We disagree with the notion of including additional antigens being a potential revolutionary improvement, in and of itself. Both BioThrax and AVP contain a mix of several B. anthracis antigens. The statement referenced in the summary notes the need for single dose vaccines, or other means to improve operational logistics. We do agree that a multi-antigen approach could be part of meeting that goal, but given the experiences with BioThrax and AVP, it seems likely that it would be in combination with the approaches in lines 190-199.

Formatting of references is inconsistent.  It seems this was not done with a bibliographic manager, e.g. Endnote, or the program was not configured to handle documents from sources other than journals. Thank you for catching. There were errors in references 4-8, 15, and 29. Those have been corrected.

Ref. 8 and 9 are book chapters.  Reviews in more easily accessible journals would be of more value to readers. We agree about the accessibility but the original sources looked to be the book chapters. Please let us know if the editors feel the need to provide references based on accessibility.

Ref. 8:   “membrane”, not “membrain” This was corrected.

Line 90:   intalize B. anthracis. Thank you, this has been corrected.

Line 93.  Antibiotic, not antibody. Thank you for catching, this was corrected to antibiotic.

Line 233:   anthrax. This has been corrected.

Line 262:   clinical. This has been corrected.

 Line 271:   spacing lost. The spacing was corrected to be consistent with the other references.